

# The complete genome of *Banana streak GF virus* Yunnan isolate infecting Cavendish *Musa* AAA group in China

Wei-li Li[1,*], Nai-tong Yu[1,2,*], Jian-hua Wang[1], Jun-cheng Li[3] and Zhi-xin Liu[1,2]

[1] Key Laboratory of Biology and Genetic Resources of Tropical Crops, Ministry of Agriculture and Rural Affairs, Institute of Tropical Bioscience and Biotechnology, Chinese Academy of Tropical Agricultural Sciences, Haikou, China

[2] Hainan Key Laboratory of Tropical Microbe Resources, Haikou, China

[3] Guangdong Key Laboratory of Tropical and Subtropical Fruit Tree Research, Institute of Fruit Tree Research, Guangdong Academy of Agricultural Sciences, Guangzhou, China

[*] These authors contributed equally to this work.

Corresponding authors
Nai-tong Yu, yunaitong@163.com
Zhi-xin Liu, liuzhixin@itbb.org.cn

## ABSTRACT

*Banana streak virus* (BSV) belongs to the members of the genus *Badnavirus*, family *Caulimoviridae*. At present, BSV contains nine species in the International Committee on Taxonomy of Viruses (ICTV) classification report (2018b release). Previous study indicated that the viral particles of *Banana streak virus Acuminata Yunnan* (BSV-Acum) were purified from banana (Cavendish *Musa* AAA group) leaves in Yunnan Province, China, and its complete genome was obtained. To further determine whether this sample infecting with *Banana streak GF virus* (BSGFV), the polymerase chain reaction (PCR) cloning and complete genome analysis of the *Banana streak GF virus* Yunnan isolate (BSGFV-YN) isolate were carried out in this study. The result showed that BSGFV-YN infecting Cavendish *Musa* AAA group was co-infecting this sample. Its genome contains a total of 7,325 bp in length with 42% GC content. This complete genome sequence was deposited in GenBank under accession number MN296502. Sequence analysis showed that the complete genome of BSGFV-YN was 98.14% sequence similarity to BSGFV Goldfinger, while it was 49.10–57.09% to other BSV species. Two phylogenetic trees based on the complete genome and ORFIII polyprotein indicated that BSGFV-YN and other BSV species clustered into a group, while it was the highest homology with BSGFV Goldfinger. Although BSGFV-YN and BSGFV Goldfinger were highly homologous, their cultivating bananas are different. The former cultivating banana was from Cavendish *Musa* AAA group, while the latter cultivating banana was from Goldfinger *Musa* AAAB group. Compared with BSGFV Goldfinger, the genome of BSGFV-YN has an extra multiple repetitive sequences in the intergenetic region between *ORFIII* and *ORF*I, suggesting that this region might be related to host selection. In summary, a BSGFV-YN distant from BSV-Acum was identified from the same sample, and its complete genome sequence was determined and analyzed. The study extends the polymorphism of BSVs in China and provides scientific clue for the evolutionary relationship with host selection of badnaviruses.

## INTRODUCTION

Banana (*Musa spp.*), a perennial monocotyledonous herb, is the fourth largest food crops and the third largest tropical fruit in the world. Bananas are susceptible to variety of viruses (*Gaur et al., 2016*), such as *Banana streak virus* (BSV) (*Geering, Parry & Thomas, 2011*), *Banana bunchy top virus* (BBTV) (*Yu et al., 2012*) and *Cucumber mosaic virus* (CMV) (*Khaled, Wardany & Mahmoud, 2016*). Banana production is threatened by the Banana streak disease (BSD), and its pathogen belongs to the genus *Badnavirus,* family *Caulimoviridae* (*Alangar, Thomas & Ramasamy, 2016*). BSV is widely distributed in the main planting areas of banana industry in Southeast Asia and Africa, and it had seriously affected the yield and quality of bananas resulted in huge economic losses (*Kumar et al., 2015*). Moreover, BSV genome may integrate into the banana genome, and it can be activated to produce infectious virions under certain environmental stress (*Gayral et al., 2008*; *Côte et al., 2010*). BSV is a kind of pararetroviruses (EPRVs) that use a virus-encoded reverse transcriptase (RT) to reverse viral RNA (vRNA) into viral DNA, completing the viral DNA replication process (*Hohn & Rothnie, 2013*). BSV possesses an open-circular double-stranded DNA genome of 7–8 kb in size and its genome is encapsidated inside non-enveloped bacilliform particle (30 nm × 150 nm) (*Selvarajan, Balasubramanian & Gayathrie, 2016*; *Alangar, Thomas & Ramasamy, 2016*).

The genomic structure of the typical badnavirus consists of three open reading frames (ORFs) in the positive strand (*Vo, Campbell & Mahfuzc, 2016*). *ORF*I and *ORF*II encode proteins of unkown function. *ORF*III, the largest ORF, encodes a ~200 kDa polyprotein which is hydrolyzed by protease to produce a variety of functional proteins related to the viral life cycle, containing movement protein (MP), coat protein (CP), aspartic protease (AP), reverse transcriptase (RT) and ribonuclease H (RNaseH) (*Duroy et al., 2014*; *Alangar, Thomas & Ramasamy, 2016*). Bioinformatic analysis indicated that the sequence between RT and RNaseH (RT/RNaseH) are the most conserved region in the badnavirus genome. According to the classification of the genus *Badnavirus* by the International Committee on Taxonomy of Viruses (ICTV), the nucleotide sequence similarity less than 80% or the amino acid sequence similarity less than 89% is considered as a new species (*Geering et al., 2014*). At present, nine BSV species of *Banana streak GF virus* (BSGFV), *Banana streak IM virus* (BSIMV), *Banana streak MY virus* (BSMYV), *Banana streak OL virus* (BSOLV), *Banana streak UA virus* (BSUAV), *Banana streak UI virus* (BSUIV), *Banana streak UL virus* (BSULV), *Banana streak UM virus* (BSUMV) and *Banana streak VN virus* (BSVNV) are identified by ICTV. In addition, three other BSV species of *Banana streak CA virus* (BSCAV) (*James et al., 2011*), *Banana streak virus-isolate GD* (BSV-GD) (*He et al., 2009*) and *Banana streak virus* Acuminata Yunnan (BSV-Acum) (*Zhuang et al., 2011*) have not been classified.

The diversities of complete genome sequences of more than nine BSV species indicated that the virus is highly variable and polymorphic (*Iskra-Caruana et al., 2014*). Furthermore, it is difficult to study the invasion mechanism and pathogenesis, owing to the instability of symptoms on the host and the integration of the BSV genome into host genome which can be activated to produce infectious virions under certain environmental stress (*Stainton et al., 2015*). The genomic characters and sequence diversity of *Banana streak virus* (BSV) in
China are under investigation. Isolation and sequencing of the BSGFV and other new BSVs are greatly abundant the diversity of the badnavirus in China and provides an important data for disease resistance breeding. Based on the characteristics of circle double-stranded DNA molecule of the BSV genome, the complete genome sequence of BSGFV-YN was obtained by segmental PCR amplification, and the genomic structure and evolutionary relationship were further analyzed. The study will extend the polymorphism of BSV in China and provide scientific clue for the evolutionary relationship with host selection of badnaviruses.

## MATERILS AND METHODS

### Materials

Banana leaves showing streak symptoms were collected by permissions of a farmer (Shao-cheng Shen) from Yunnan, China in 2009. The banana belongs to the Cavendish *Musa* AAA group. Total DNA was extracted and stored at −80 °C. Previous study showed that the sample was infected with *Banana streak virus* Acuminata Yunnan (BSV-Acum), and its complete genome was obtained and analyzed in our laboratory (*Zhuang et al., 2011*). In order to further determine whether other BSV species or strains co-infect in the same plant, the complete genome amplification based on PCR method was conducted on the total DNA.

### Cloning and sequencing of viral genome

According to the complete genome sequence of BSGFV deposited in NCBI (GenBank accession number NC_007002.1), seven specific pairs primers were designed by Primer Premier 5.0 (Table 1). The primers were synthesized by Invitrogen (Guangzhou, China) Trading Co., Ltd. The PCR reaction system (50 µL) contained *EasyTaq®* DNA Polymerase 1 µL (Beijing, China), 2.5 mM dNTPs 4 µL, 10 ×*EasyTaq* buffer 5 µL, 5 µM forward and reverse primers 4 µL, total DNA template 2 µL, and added ddH$_2$O to 50 µL. PCR reaction condition was pre-denaturation at 94 °C for 3 min; followed by 35 cycles of denaturation at 94 °C for 30 s, annealing at 55 °C for 30 s, and extension at 72 °C for 90 s; and a final extension step at 72 °C for 10 min. The 6 µL PCR product was analyzed by electrophoresis in 1.5% agarose gel, and the remaining PCR product was purified by an OMEGA gel recovery kit (Bio-Tek, USA). The purified PCR product of each DNA fragment was cloned into the pMD18-T vector (Takara, China), and then transformed into *E. coli* DH5a competent cells (2nd Lab, Shanghai, China). Three positive clones were randomly selected for Sanger sequencing at Invitrogen (Guangzhou, China).

### Complete genome sequence assembly and analysis

The seven overlapping fragments were edited by ChromasPro software (Technelysium Pty. Ltd., Australia) and were used to assemble into the complete genome of BSGFV-YN by BioEdit software (*Hall, 1999*). The genome was further analyzed by Blastn and Blastx at NCBI website (http://blast.ncbi.nlm.nih.gov/Blast.cgi), and the possible species or strain of BSV was preliminarily identified. Based on the classification of ICTV (2018b release), 37 badnaviruses were downloaded from GenBank for further comparison and analysis

**Table 1** Primers used in this study.

| Primer | Primer sequence (5′ to 3′ direction) | Position (nt) | Direction |
|---|---|---|---|
| ORFI-F | ATGAACTCCGACCTCAAAGAG | 484–504 | Forward |
| ORFI-R | TCCAAGAATTGTTTTTCTTAGATGATG | 982–1008 | Reverse |
| ORFII-F | ATGAACTCAGAGGCATACAAGG | 1008–1029 | Forward |
| ORFII-R | TTGAATCTCCTTGAGAAGGTCAAAAG | 1318–1343 | Reverse |
| P1-F | CGGACGTGGTGGACCCAGC | 6844–6862 | Forward |
| P1-R | CCGAAGGTTGTGAGCTAAGTCAG | 590–612 | Reverse |
| P2-F | GAACCCTTTTGACCTTCTCAAG | 1313–1334 | Forward |
| P2-R | TTGTCCCATCTTTTGGGCTTCC | 2751–2772 | Reverse |
| P3-F | GGGAGCTCTCGGGGTTG | 2699–2715 | Forward |
| P3-R | GGCAGAACTTCCTTAGTAGTTCG | 4108–4130 | Reverse |
| P4-F | GCATGACTGGAGAGAACTAGC | 4075–4095 | Forward |
| P4-R | TACCTCGGAAGCAGTTGTCCAT | 5504–5525 | Reverse |
| P5-F | GATGGACAACTGCTTCCGAGG | 5503–5523 | Forward |
| P5-R | ACACGCTAGTATGTGCTGGC | 6967–6986 | Reverse |

(Table 2). The open reading frames (ORFs) of BSGFV-YN were predicted by ORF Finder online website (https://www.ncbi.nlm.nih.gov/orffinder/) and bioinformatic analysis. Subsequently, the sequence similarities of complete genomes, *ORF*I~III and their putative proteins between BSGFV-YN with other 15 badnaviruses were analyzed by Sequence Identity Matrix Program in BioEdit software. The movement protein (MP), coat protein (CP), aspartic protease (AP), reverse transcriptase (RT), ribonuclease H (RNaseH) and two cysteine-rich domains in BSGFV-YN *ORF*III were predicted according to the report by *Li et al. (2018)*, and sequence similarities with other homologous proteins were aligned by the GeneDoc software. The repeated sequences in the intergenic regions between *ORF*III and *ORF*I of BSGFV-YN and BSGFV goldfinger (NC_007002.1) were also analyzed by GeneDoc software.

## Phylogenetic analysis

In order to further determine the homology and evolutionary relationships of BSGFV-YN with other badnaviruses, two phylogenetic trees based on the complete genomes and ORFIII polyproteins of BSGFV-YN and other 37 badnaviruses were constructed by using MEGA6.0 software (Table 2). The evolutionary history was inferred using the Neighbor-Joining method and its distance was computed using the p-distance method (*Saitou & Nei, 1987*). Meanwhile, to analyze the evolutionary relationship between BSGFV-YN and other BSV species in different banana hosts, a phylogenetic tree was also constructed based on the complete genome sequences.

## RESULTS

### Cloning and assembly of BSGFV-YN genome

To obtain the complete genome sequence of BSGFV-YN, 7 specific pair primers were used for segmental PCR amplification (Fig. 1A). The PCR results showed that each DNA

**Table 2  The information of 38 badnaviruses used in this study.**

| Virus | Abbreviation | Genbank no. | Host |
|---|---|---|---|
| *Aglaonema bacilliform virus* | ABV | MH384837.1 | *Aglaonema commutatum* |
| *Banana streak GF virus* Yunnan isolate | BSGFV-YN | MN296502 | *Musa* AAA group |
| *Banana streak GF virus* | BSGFV | NC_007002.1 | Goldfinger (AAAB) |
| *Banana streak IM virus* | BSIMV | NC_015507.1 | *Musa* sp. cv. Mshule |
| *Banana streak MY virus* | BSMYV | KJ013509.1 | *Musa* acuminata cv. Cavendish |
| *Banana streak OL virus* | BSOLV | NC_003381.1 | *Musa* AAA group |
| *Banana streak UI virus* | BSUIV | NC_015503.1 | *Musa* sp. cv. Kisansa |
| *Banana streak UL virus* | BSULV | NC_015504.1 | *Musa* sp. cv. Kibuzi |
| *Banana streak UM virus* | BSUMV | NC_015505.1 | *Musa* sp. cv. Mbwazirume |
| *Banana streak VN virus* | BSVNV | KJ013510.1 | *Musa* acuminata cv. Cavendish |
| *Banana streak UA virus* | BSUAV | NC_015502.1 | *Musa* sp. cv. Likhako |
| *Banana streak virus* Acuminata Yunnan | BSV-Acum | NC_008018.1 | plaintain |
| *Birch leaf roll-associated virus* | BLRaV | NC_040635.1 | *Betula pubescens* |
| *Blackberry virus F* | BbVF | NC_029303.1 | Blackberry |
| *Bougainvillea chlorotic vein banding virus* | BCVBV | NC_011592.1 | *Chgainvillea spectabilis* |
| *Cacao mild mosaic virus* | CMMV | NC_033738.1 | Theobroma cacao |
| *Cacao bacilliform Sri Lanka virus* | CBSLV | MF642736.1 | Theobroma cacao |
| *Cacao yellow vein banding virus* | CYVBV | NC_033739.1 | Theobroma cacao |
| *Cacao swollen shoot virus* | CSSV | NC_001574.1 | Theobroma cacao |
| *Commelina yellow mottle virus* | ComYMV | NC_001343.1 | *Commelina* |
| *Canna yellow mottle associated virus* | CaYMV | KY971493.1 | Canna sp. |
| *Citrus yellow mosaic virus* | CYMV | EU708317.1 | Acid lime |
| *Dioscorea bacilliform virus* | DBV | NC_009010.1 | *Dioscorea sansibarensis* |
| *Fig badnavirus 1* | FBV-1 | NC_017830.1 | Fig |
| *Grapevine vein clearing virus* | GVCV | NC_015784.2 | Grapevine (*Chardonel*) |
| *Gooseberry vein banding associated virus* | GVBaV | NC_018105.1 | *Ribes rubrum* cv. Holandsky cerveny |
| *Grapevine Roditis leaf discoloration-associated virus* | GRLDaV | NC_027131.1 | *Vitis vinifera* |
| *Jujube mosaic-associated virus* | JMaV | NC_035472.1 | Ziziphus jujube |
| *Kalanchoe top-spotting virus* | KTSV | NC_004540.1 | *Kalanchoe blossfeldiana* |
| *Pineapple bacilliform CO virus* | PBCOV | NC_014648.1 | Pineapple plants |
| *Piper yellow mottle virus* | PYMV | NC_022365.1 | *Piper nigrum* |
| *Pagoda yellow mosaic associated virus* | PYMAV | NC_024301.1 | *Styphnolobium japonicum* (L.) Schott |
| *Rubus yellow net virus* | RYNV | KM078034.1 | Rubus sp. |
| *Sugarcane bacilliform MO virus* | SBMOV | NC_008017.1 | Sugarcane |
| *Sweet potato pakakuy virus* | SPPV | NC_015655.1 | Sweet potato |
| *Taro bacilliform virus* | TaBV | NC_004450.1 | Taro |
| *Wisteria badnavirus 1* | WBV-1 | NC_034252.1 | *Wisteria sinensis* |
| *Yacon necrotic mottle virus* | YNMV | NC_026472.1 | Yacon |

fragment was specifically amplified and the length was consistent with the expected size. Bioinformatic analysis showed that the sequence of each fragment was highly similar to the BSGFV by blastn. The seven overlapping DNA fragments were assembled via head-to-tail method by BioEdit software, and a circular double-stranded DNA of 7,325 bp with 42%

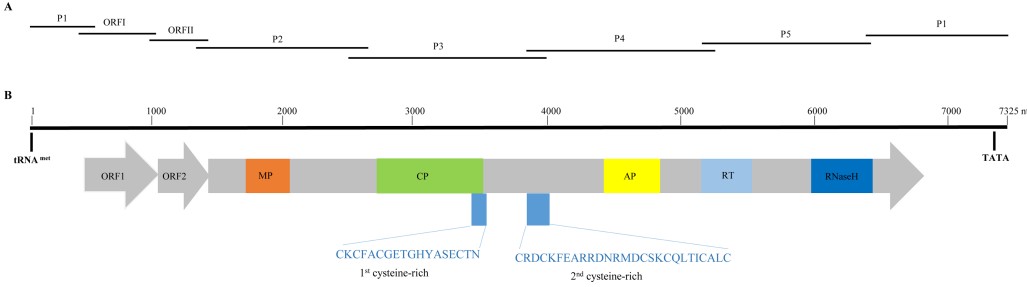

**Figure 1 Overlapping PCR amplification and schematic genome organization of *Banana streak GF virus* YN isolate (BSGFV-YN).** (A) The complete genome of BSGFV-YN was amplified by using seven overlapping PCR of P1, ORF I, ORF II, P2, P3, P4 and P5. (B) The putative three open reading frames (ORFs) are indicated and the predicted domains of movement protein (MP), coat protein (CP), aspartic protease (AP), reverse transcriptase (RT) and Ribonuclease H (RNase H) within *ORF III* are shown. Numbers above indicate the total number of nucleotide sequences of the BSGFV-YN. The amino acid sequences of 1st cysteine-rich and 2nd cysteine-rich regions were also indicated.

GC was finally obtained. The complete genome sequence of BSGFV-YN was deposited in GenBank under accession number MN296502.

## Genomic analysis of BSGFV-YN

Sequence analysis showed that the characteristic of BSGFV-YN complete genome was similar to the typical badnavirus. In detail, the BSGFV-YN contains three open reading frames (ORFs) on the positive strand of viral genome. The first ORF (*ORF*I) was found at 484–1,011 bp while the second ORF (*ORF*II) was located at 1,008–1,346 bp, both of which were predicted to encode proteins with unknown function. However, the third ORF (*ORF*III) located at 1,343-6,841 bp encoded a polyprotein with the largest in size (Fig. 1B). Further analysis revealed that the termination codon region of BSGFV-YN *ORF*I has a 4-base overlap (ATGA) with the start codon region of *ORF*II; similarly, the termination codon region of *ORF*II and the start codon region of *ORF*III are also overlap of a 4-base, with ATGA as well.

Complete genome sequence analysis showed that BSGFV-YN was 98.14% sequence similarity with BSGFV Goldfinger, while it was 49.10–57.09% sequence similarity with other BSV species. The genome size of BSGFV is the smallest among all BSV species. In addition, the sequence similarity of BSGFV-YN with *Canna yellow mottle associated virus* (CaYMV), *Kalanchoe top-spotting virus* (KTSV), *Pineapple bacilliform CO virus* (PBCOV), or *Sugarcane bacilliform MO virus* (SBMOV) was 53.58%, 52.92%, 51.49% and 49.16%, respectively. Further, the BSGFV-YN ORFs (I~III) were 99.43%, 99.11% and 98.91% nt sequence similarities with those of BSGFV Goldfinger, while they shared 99.62%, 98.82% and 99.02% at the amino acid levels. The sequence similarity between BSGFV-YN ORFs (I~III) and other badnaviruses ORFs (I~III) were even more lower (Table S1).

*Badnavirus ORF*III, the largest ORF, encodes a ~200 kDa polyprotein, which can be hydrolyzed by protease into several small proteins, such as MP, CP, AP, RT and RNase H. In this study, BSGFV-YN ORFIII polyprotein also contains MP, CP, AP, RT, RNaseH and two cysteine-rich zinc finger-like RNA-binding regions (Figs. 1B and 2). The domains of

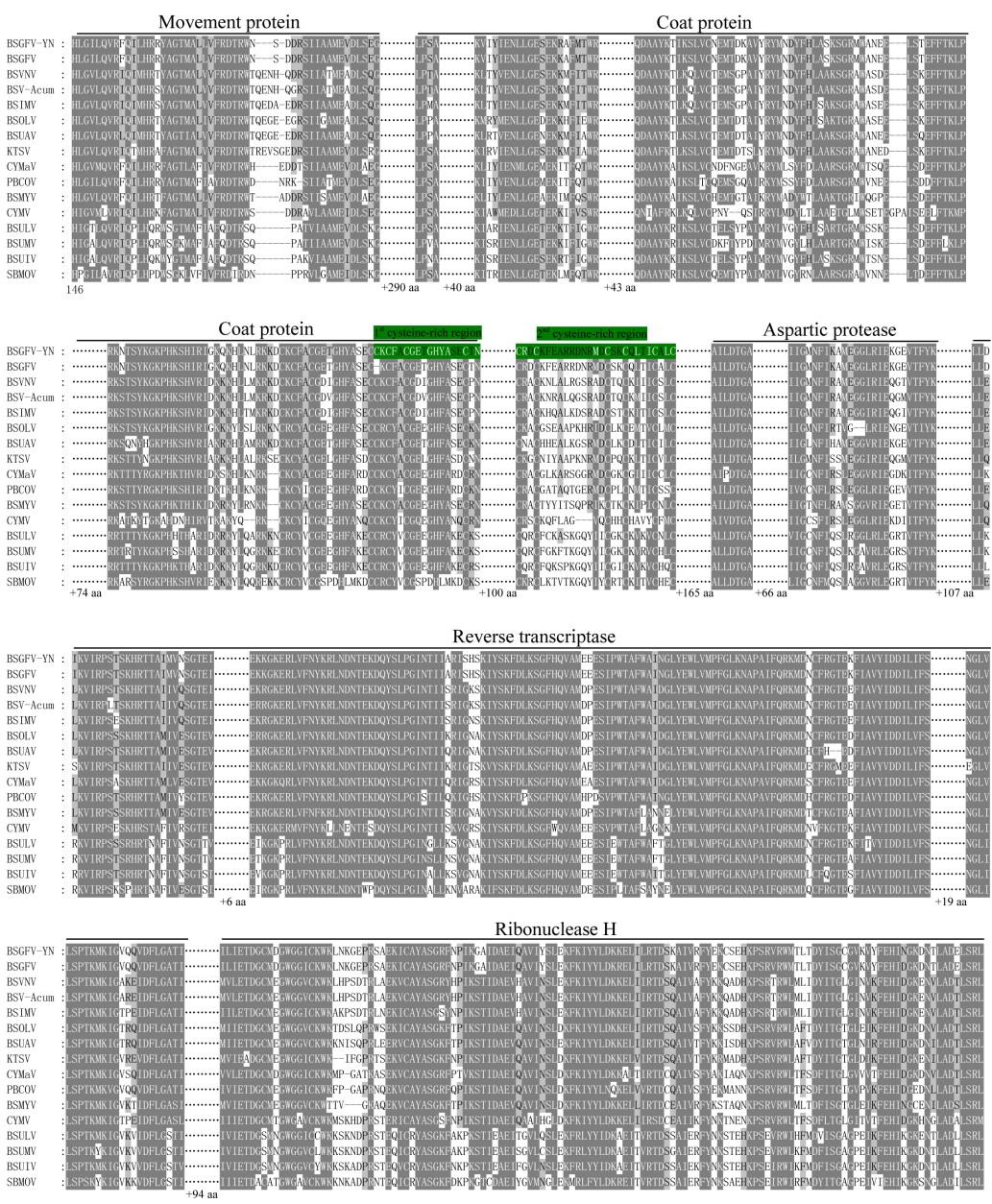

**Figure 2** Comparison of amino acid sequences of the conserved domains in the putative ORFIII polyprotein of BSGFV-YN with those of representative badnaviruses. Comparison of amino acid sequences of the conserved domains in the putative ORFIII polyprotein of BSGFV-YN with those of representative badnaviruses. The predicted domains of movement protein (MP), coat protein (CP), aspartic protease (AP), reverse transcriptase (RT) and Ribonuclease H (RNase H) within ORFIII are shown. Two cysteine-rich regions are highlighted with green color.

BSGFV-YN ORFIII are highly similar or identical to the corresponding domains of BSGFV Goldfinger. Further analysis showed that the RT domain was the most conservative with the similarity of 72.25~84.29%, while the AP was the most variable, with the similarity of 39.39~66.33% (Table S2).

```
                           Repeat S                                Repeat L
BSGFV-YN : AGCACATACTGGCGTGTAAAGGCATCTGGTTGTCCCCAGAAGGCCTAAAGTTAGTGCGTTCCAACGCACATCTGCGTGTA :  7049
BSGFV    : AGCACATACTAGCGTGTAAAGGCATCTGGTTGTCCCCAGAAGGCCTAAAGTTAGTGCGTTCCAAC--------------- :  7049

           Repeat S                        Repeat L                         Repeat S
BSGFV-YN : AAGGCATCTGGTTGTCCCCAGAAGGCCTAAAGTTAGTGCGTTCCAACGCACATCTGTGTGTAAAGGTATCTGGCTGTTTC :  7129
BSGFV    : ------------------------------------------------GCACACCTATGTGTAAAGGTATCTGGCTGTTTC :  7129

BSGFV-YN : CAGACGCTACCTCCCTCTTT :  7149
BSGFV    : CAGACGCTACCTCCCTCTTT :  7149
```

**Figure 3** **Comparison of non-coding region sequences of the BSGFV-YN with BSGFV Goldfinger.** Comparison of non-coding region sequences of the BSGFV-YN with BSGFV Goldfinger. The repeat sequences are indicated in different color. Three ''Repeat S'' sequences (red color) and two ''Repeat L'' sequences (blue color) are shown.

The non-coding region of BSGFV-YN genome is 968 bp, which contains a specific binding site, the 12 bases (*5′-TGG TAT CAG AGC-3′*), of tRNA[met]. This binding site is a common feature among all badnaviruses. The first nucleotide ''T'' is a highly conserved base required for replication and is set as the starting base of the sequence. The non-coding region also contains a TATA box domain (TATATA), which is located at 7,162–7,168 bp. In addition, the non-coding region has a 63 bp sequence that longer than BSGFV Goldfinger, and two kinds of repeating sequences were found in the 63 bp and adjacent region. A short repetition ''Repeat S'' and relative long repetition ''Repeat L'' have been found (Fig. 3).

## Phylogenetic analysis

In order to analyze the evolutionary relationships between BSGFV-YN and 37 other badnaviruses, a phylogenetic tree based on their complete genomes was firstly constructed. The results showed that these badnaviruses can be divided into two distinct groups, Group I and Group II. BSGFV-YN and other BSV species were classified into the Group II. Furthermore, 11 BSVs can be further divided into two subgroups, BSV-I and BSV-II subgroups. BSGFV-YN was the highest homology with the BSGFV Goldfinger, but it was more close to KTSV than other BSV species (Fig. 4). Similar result has been reported in badnaviruses (*Rumbou et al., 2018*).

To further determine the phylogenetic relationship, another phylogenetic tree was constructed based on ORFIII polyproteins. The results confirmed that these badnaviruses can be divided into Group I and Group II. However, the *Aglaonema bacilliform virus* (ABV) group was clustered into Group I, which is different from the result of phylogenetic tree based on the complete genome. This phylogenetic tree also showed that BSGFV-YN had the highest homology with BSGFV Goldfinger (Fig. 5).

As shown in Fig. 6, phylogenetic tree was constructed by the eleven complete genomes of BSVs. The results showed that BSGFV-YN and other BSV species formed to two branches. In detail, BSUMV, BSUIV and BSULV formed to one branch, while BSGFV-YN, BSGFV Goldfinger, BSIMV, BSMYV, BSOLV, BSVNV, BSUAV, and BSV-Acum were clustered into another branch. The BSGFV-YN was the highest homology with BSGFV Goldfinger, which was consistent with results as Figs. 4 and 5A. It is interesting that BSGFV Goldfinger

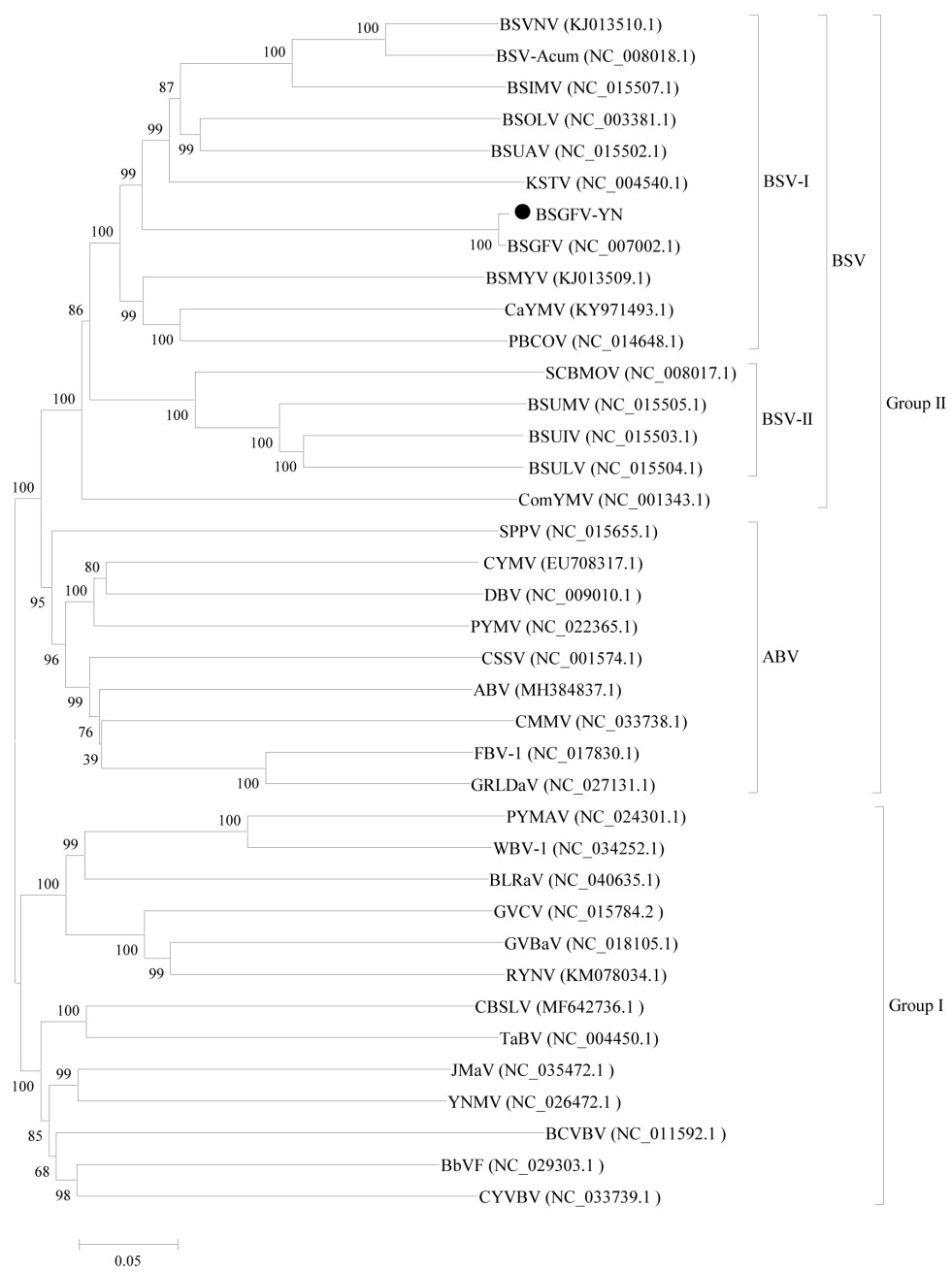

**Figure 4  Phylogenetic tree of the complete genome sequences of BSGFV-YN and other members of** *Badnavirus* **were conducted in MEGA6.** Phylogenetic tree of the complete genome sequences of BSGFV-YN and other members of *Badnavirus* were conducted in MEGA6. The evolutionary history was inferred using the Neighbor-Joining method and its distances were computed using the p-distance method. The percentage of replicate trees in which the associated taxa clustered together in the bootstrap test (1,000 replicates) is shown next to the branches. The analysis involved 38 complete genome sequences in *Badnavirus*. All positions containing gaps and missing data were eliminated. There were a total of 3,805 positions in the final dataset.

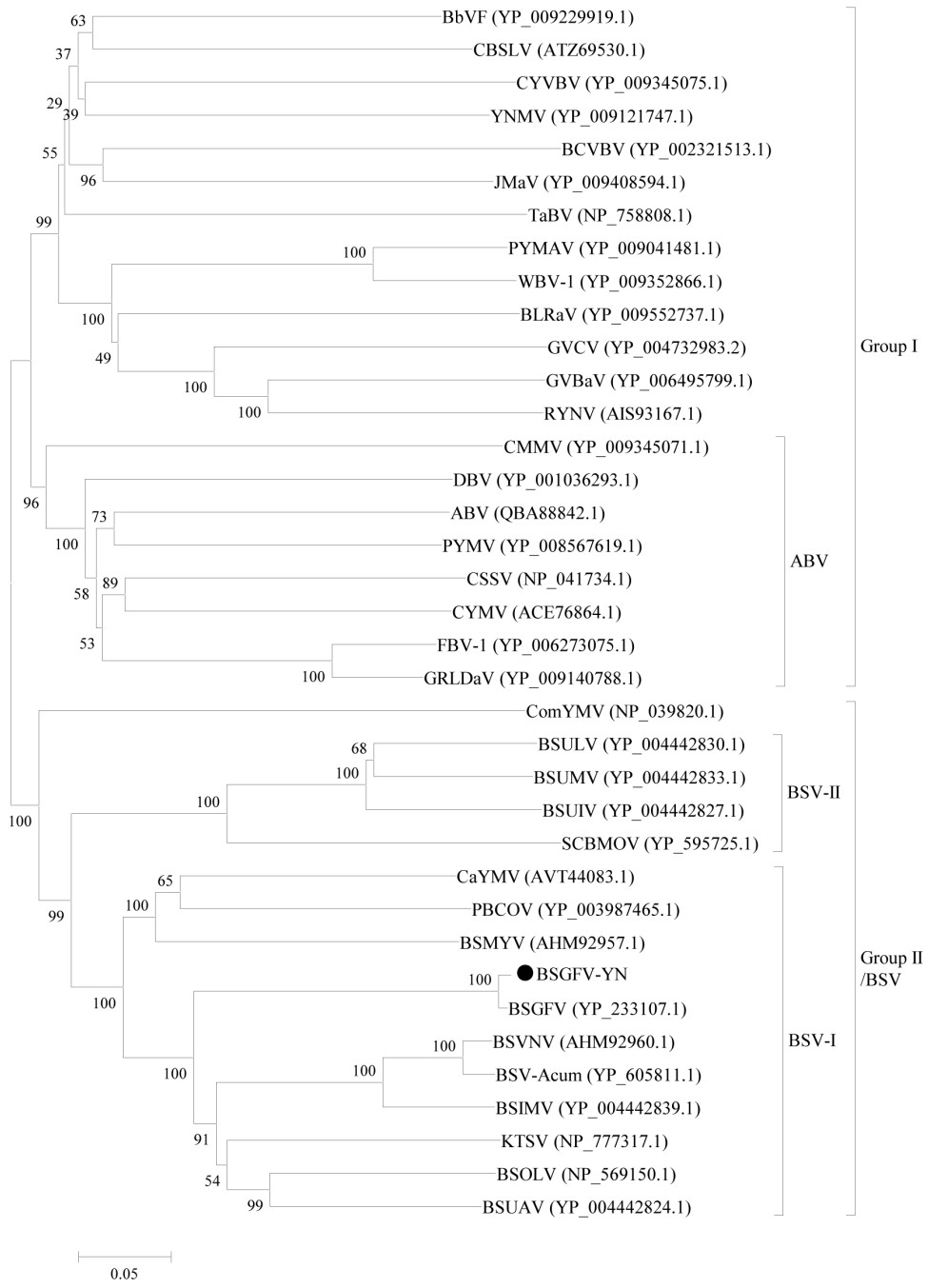

**Figure 5  Phylogenetic tree of the amino acid sequences of BSGFV ORFIII and other ORFIIIs of the members in *Badnavirus* were conducted in MEGA6.** Phylogenetic tree of the amino acid sequences of BSGFV ORFIII and other ORFIIIs of the members in *Badnavirus* were conducted in MEGA6. The evolutionary history was inferred using the Neighbor-Joining method and the evolutionary distances were computed using the p-distance method. The percentage of replicate trees in which the associated taxa clustered together in the bootstrap test (1,000 replicates) is shown next to the branches. The analysis involved 37 ORFIII amino acid sequences. All positions containing gaps and missing data were eliminated. There were a total of 1,117 positions in the final dataset.

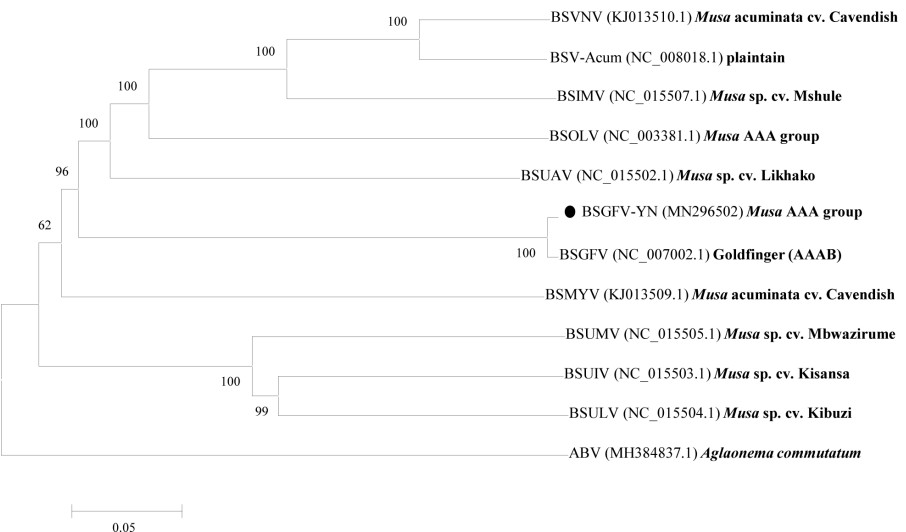

**Figure 6** **Phylogenetic tree of the complete genome sequences of BSGFV-YN and other BSV isolates were conducted in MEGA6.** Phylogenetic tree of the complete genome sequences of BSGFV-YN and other BSV isolates were conducted in MEGA6. The evolutionary history was inferred using the Neighbor-Joining method and the evolutionary distances were computed using the p-distance method. The percentage of replicate trees in which the associated taxa clustered together in the bootstrap test (1,000 replicates) is shown next to the branches. The analysis involved 12 nucleotide sequences. All positions containing gaps and missing data were eliminated. There were a total of 6,121 positions in the final dataset. Bold fonts are the host names of the viruses.

has been reported to infect Goldfinger (*Musa* AAAB), but the BSGFV-YN was cloned from another banana line (*Musa* AAA), indicating the BSGFV can infect different banana varieties.

## DISCUSSION

Currently, fifty-seven members are recognized in the genus *Badnavirus* of the family *Caulimoviridae*, including nine species of BSV, suggesting the high diversity and polymorphism of BSV species. The badnaviruses, one of the important plant viral pathogens, endanger global agriculture. They are widely infecting the economic crops, such as bananas, sugarcane, citrus, cocoa, taro, black pepper, yam and other tropical and subtropical crops resulting in severe diseases (*Lee et al., 2015*; *Alangar, Thomas & Ramasamy, 2016*). The badnaviruses have been concerned by people recently because some of badnaviruses have been causing significant agricultural economy loss. Among them, BSV is widely distributed in the main planting areas of banana industry (*Jaufeerally, Khorugdharry & Harper, 2006*), and its genome can be integrated into the banana host genome which can be activated to produce infectious virions under certain environmental stress (*Umber et al., 2016*; *Chabannes et al., 2013*). Based on this situation, the potential endanger by BSV is immeasurable. Therefore, it is very necessary to recognize the genomic structure and polymorphism of viral genome, protein function, virus evolution,

geographical distribution, host range and pathogenic mechanism of BSV species (or strains) on different banana lines.

In this study, the complete genome of the BSGFV-YN (GenBank accession MN296502) was obtained by seven segmental PCR cloning strategies. Previous study has been showed that the sample was infected with *Banana streak virus* Acuminata Yunnan (BSV-Acum), and its complete genome was obtained from the purified viral particle (*Zhuang et al., 2011*). Similarly, the BSGFV-YN can be amplified from the viral particle and its genome is episomal in the host banana. BSGFV-YN has a typical genomic characteristic of badnavirus and phylogenetic tree analysis indicated that it is highly homologous to the BSGFV Goldfinger. According to the classification of the ICTV, the amino acid sequence similarity of the RT/RNase H less than 89% can be determined as a new species. In this study, the RT and RNase H of BSGFV-YN were 100% and 98.45% similar to the reported BSGFV Goldfinger, indicating that BSGFV-YN and BSGFV Goldfinger belong to the same BSV species. Although they are highly homologous, their hosts are obviously different. The former banana host is Cavendish *Musa* AAA group while the latter banana host is Goldfinger *Musa* AAAB group. We found that BSGFV-YN had an extra specific sequence longer than BSGFV Goldfinger between the intergenic region of *ORF*III and *ORF*I. Sequence analysis indicated that two kinds of repeated sequence were found. Studies have been shown that BSOLV (NC_003381) was 441 bp longer than BSOLV-IN1/IN2 between the regions of the *ORF*III and *ORF*I, but their host derived from the different banana varieties (*Baranwal et al., 2014*). The evidence indicates that the repeated sequence of the intergenic region may provide an important clue for the BSGFV infecting different banana hosts, but further experimental evidences are needed.

This study showed that RT was the most conserved domain in the polyprotein, while AP was the lowest conserved. The badnaviruses firstly transcribed genome DNA into the full-length viral RNA (vRNA), and then reverse transcribed to viral DNA by RT to complete viral DNA replication. Therefore, the RT enzyme is conserved in all badnaviruses, which is consistent with its protein function (*Li et al., 2018*). AP, as a proteolytic enzyme, can hydrolyze the polyprotein into several functional small proteins. Depending on its specific proteolytic sites, we suggested that the conservation of AP is low (*Hany et al., 2014*; *Vo, Campbell & Mahfuzc, 2016*).

## CONCLUSIONS

In summary, BSGFV-YN, a new isolate of the *Banana streak GF virus* (BSGFV), was cloned from the banana sample of Cavendish *Musa* AAA group that has been reported as being infected with BSV-Acum. Bioinformatics analysis showed that BSGFV-YN is highly homologous to the BSGFV Goldfinger isolated from M*usa* AAAB group, indicating that BSGFV can infect different banana varieties. BSGFV-YN may be a novel isolate of BSGFV in the new host, which provides an important clue for revealing the polymorphism of BSV genome.

### Funding

This study was funded by the Young Elite Scientists Sponsorship Program by CSTC (Project No. CSTC-QN201704) and Guangdong Key Laboratory of Tropical and Subtropical Fruit Tree Research, Institute of Fruit Tree Research, Guangdong Academy of Agricultural Sciences (No. 2017B030314113). The funders had no role in study design, data collection and analysis, decision to publish, or preparation of the manuscript.

### Grant Disclosures

The following grant information was disclosed by the authors:
Young Elite Scientists Sponsorship Program by CSTC: CSTC-QN201704.
Guangdong Key Laboratory of Tropical and Subtropical Fruit Tree Research, Institute of Fruit Tree Research, Guangdong Academy of Agricultural Sciences: 2017B030314113.

### Competing Interests

The authors declare there are no competing interests.

### Author Contributions

- Wei-li Li performed the experiments, analyzed the data, prepared figures and/or tables, and approved the final draft.
- Nai-tong Yu conceived and designed the experiments, analyzed the data, prepared figures and/or tables, authored or reviewed drafts of the paper, and approved the final draft.
- Jian-hua Wang and Jun-cheng Li performed the experiments, authored or reviewed drafts of the paper, and approved the final draft.
- Zhi-xin Liu conceived and designed the experiments, analyzed the data, authored or reviewed drafts of the paper, and approved the final draft.

### Field Study Permissions

The following information was supplied relating to field study approvals (i.e., approving body and any reference numbers):

No field permits are available because authorizations were given by a farmer, Shao-cheng Shen.

### Data Availability

The complete genome sequence is available in GenBank: MN296502.

### Supplemental Information

Supplemental information for this article can be found online at http://dx.doi.org/10.7717/peerj.8459#supplemental-information.

## PeerJ

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
