# Peer review of "The complete genome of Banana streak GF virus Yunnan isolate infecting Cavendish Musa AAA group in China"

_PeerJ, doi:10.7717/peerj.8459_

## Round 0.1 · original submission · Major Revisions

Both referees liked your work but have raised issues that need to be addressed before I can accept your manuscript.

I would like you consider each one of the referees' comments as they are all important. Addressing them will help to improve your manuscript substantially. The reviewers have given you many hints to help you on this task. Please edit your manuscript so all points have been answered properly and send a point by point answer to the referees comments along with your resubmitted manuscript.

Importantly, please have the manuscript checked by a professional native English speaker. I also ask for an improvement of the quality of the figures and that you can make sequencing data available.

Thank you very much

l look forward to receiving your manuscript

Reviewer 1 ·

Basic reporting

In this manuscript the authors report a BSGFV Goldfinger strain (BSGFV-YN) infecting Cavendish banana. This virus was isolated from a previously reported infected banana with other badnavirus (BSv-Acum), a very interesting finding in my point of view, since this result supports the hypothesis that coinfection with badnaviruses is more frequent than we would expect, as suggested by some recent publications (e.g. Development of Quantitative Real-Time PCR Assays for Rapid and Sensitive Detection of Two Badnavirus Species in Sugarcane). The authors sequenced the new strain and developed phylogenetic and genetic analysis showing a new intergenic region between ORFIII and ORFI, suggesting that this region could be related to host selection.

The structure of the manuscript conforms to PeerJ standars, however, figures have very low quality and is necessary to improve them for a better understanding of the manuscript.

Sequencing raw data are not supplied and I think that must be apported to validate the results. E.g. quality histograms are crucial to confirm the results obtained and subsequent phylogenies studies.

Authors must justify why they decided to analyze this Banana sample again, looking for new viruses and more specifically BSGFV. Did any phenotype that suggest a more complex infection on that?

Experimental design

I have some comments about the cloning and sequencing of viral genome. Badnaviruses are RNA viruses. As mayority of them, they have a high mutation rate and is was demonstrated that viral population holds significant haplotype diversity and primers must be designed taken in account this. Authors should indicate the criteria of primers design. Another point I disagree is the use of a routine taq polymerase to amplify the segments of the virus. I think this kind of experiment need to be performed with a high fidelity polymerase to avoid amplification errors. I observe that some amplicons are not overlapping, at this point could be interesting that authors provide the raw data to see the quality in the beginning of sequencing to be sure that there is no possibility of error in these regions.
phylogenetic studies were carried out correctly, taking into account different approaches in the phylogenetic studies of badnaviruses

Validity of the findings

As I understand, authors suggest different hosts target due to the intergenic region between ORFIII and ORFI. However, I think that this conclusion needs further experiments to be addressed. Can BSGFV Goldfinger infects Cavendish banana or BSGFV-YN infects Goldfinger banana? If one of these strain of BSGFV is able to infect those two cultivars invalidates that hypothesis. I think this point is important to know if the findings support the discovery of a new strain or an isolate in a new host.

L245- With the results shown in the manuscript it is very risky to say that BSGFV-YN is a new species of BSV. Since, from my point of view, as a result of the data provided, BSGFV-YN could be no more than an isolate of BSGFV Goldfinger in a new host.

Additional comments

Some minor comments
L28-29 The last classification report of International Committee on Taxonomy of Viruses (ICTV) is 2018b release.

L191- This result is not surprising for me. In fact, at least five species of BSV including BSV-Acum are closer to KTSV than BSGFV-YN as authors show in the fig.4. Also, there are previous reports about similar results like Rumbou A et al. (2018) A novel badnavirus discovered from Betula sp. affected by birch leaf-roll disease. PLoS ONE 13(3): e0193888. That showed similar result in phylogenetic analysis.

·

Basic reporting

The manuscript needs substantial revision for English language owing to lack of flow and presence of many grammatical errors, few of which are listed below for example:
1. Line 58: "At present, BSV widely distributed in the main planting area" should be "At present, BSV is widely distributed in the main planting area".
2. Line 58-59: "banana industry in Southeast Asia and Africa, which has seriously affecting the yield and quality of bananas and causing huge economic losses" should be "banana industry in Southeast Asia and Africa, and had seriously affected the yield and quality of bananas resulted in huge economic losses".
3. Line 63: "BSV possess" should be "BSV possesses".
4. Line 64: "and its genome encapsidated" should be "and its genome is encapsidated".
5. Line 68: "polyprotein which hydrolysis by proteases" should be "polyprotein which is hydrolyzed by proteases".
The errors are too many to list here.
The literature references are fine and cited in appropriate context. The manuscript has an acceptable format with good sectioning structure and is self-contained as well.

Experimental design

The manuscript falls under the Aims and Scope of PeerJ. The concept of the study is fine, however, investigation is not rigorous. The study presented only a single genome of Banana streak GF virus Yunnan isolate from a sample previously used to sequence another BSAcYNV isolate. There is not retaliation presented why author suspected that this sample may also have BSGFV Yunnan isolate (sequenced in this study) and why other species of BSV were not tested? It is not tested whether this isolate is episomal or integrated in the host genome?The methods are described in sufficient details for reproduction.

Validity of the findings

Currently, the manuscript is limited to sequencing of an isolate, however, no work is presented for its prevalence in the area; whether and to which extant is it present in the region? Is it present only in association with BSAcYNV or alone? Is it episomal or present as integrated in the field? The authors are encouraged to determine the presence of this isolates in the area using specific primers/PCR assay and sequence the genome of few more isolates.

---

## Round 0.2 · Minor Revisions

I will be willing to accept the manuscript once you have edited the text following the changes that reviewer 2 has clearly indicated in the Basic reporting. Thank you

Reviewer 1 ·

Basic reporting

English has been substantially improved. I consider that the references are suitable for the context of the manuscript and the results are interesting and open a new way of study to deepen the variability and distribution of viruses that infect Cavendish Musa AAA group.
It would be interesting to continue with this line in order to answer important questions such as the possible distribution of this new virus strain and its intrapopulation variability.

Experimental design

no comment

Validity of the findings

no comment

·

Basic reporting

The manuscript has been updated for English language however, few sentences needs to be changed:
1. Line 35-36: " The result showed that Banana streak GF virus Yunnan isolate (BSGFV-YN) infecting Cavendish Musa AAA group was determined " should be " The result showed that Banana streak GF virus Yunnan isolate (BSGFV-YN) infecting Cavendish Musa AAA group was co-infecting this sample ".
2. Line 46: " In summary, a BSGFV-YN different from BSV-Acum was identified from the one sample " should be " In summary, a BSGFV-YN different from BSV-Acum was identified from the same plant ".
3. Line 118: " bio-sanger sequencing at Invitrogen (Guangzhou, China). " should be " Sanger sequencing at Invitrogen (Guangzhou, China). ".
4. Line 151: " each fragment was high similar to the BSGFV " should be " each fragment was highly similar to the BSGFV ".
5. Line 210: " DISCUSS" should be " DISCUSSION".
6. Line 215 to 217: “The badnaviruses have been concerned by people recently owing to they could cause significant agricultural economy loss.” Should be rephrased for more clarity.

Experimental design

The manuscript falls under the Aims and Scope of PeerJ. The concept and experimental design of the study is fine.

Validity of the findings

In my previous review report, I have commented as “Currently, the manuscript is limited to sequencing of an isolate, however, no work is presented for its prevalence in the area; whether and to which extant is it present in the region? Is it present only in association with BSAcYNV or alone? Is it episomal or present as integrated in the field? The authors are encouraged to determine the presence of this isolates in the area using specific primers/PCR assay and sequence the genome of few more isolates”. Though the authors have not added any data for these aspects, however, their reply is logical.

---

## Round 0.3 · accepted · Accept

I see you have followed the reviewer’s minor revision recommendations so I believe you manuscript is now ready for publication. Thank you very much